# An Optimal Hierarchical Approach for Oral Cancer Diagnosis Using Rough Set Theory and an Amended Version of the Competitive Search Algorithm

**DOI:** 10.3390/diagnostics13142454

**Published:** 2023-07-24

**Authors:** Simin Song, Xiaojing Ren, Jing He, Meng Gao, Jia’nan Wang, Bin Wang

**Affiliations:** 1The Second Medical Center, Chinese People’s Liberation Army General Hospital, Beijing 100089, China; 2The First Medical Center, Chinese People’s Liberation Army General Hospital, Beijing 100853, China; xiaojingren@gmail.com

**Keywords:** diagnosis, oral cancer, rough set theory, support vector machine, K-means, amended competitive search algorithm

## Abstract

Oral cancer is introduced as the uncontrolled cells’ growth that causes destruction and damage to nearby tissues. This occurs when a sore or lump grows in the mouth that does not disappear. Cancers of the cheeks, lips, floor of the mouth, tongue, sinuses, hard and soft palate, and lungs (throat) are types of this cancer that will be deadly if not detected and cured in the beginning stages. The present study proposes a new pipeline procedure for providing an efficient diagnosis system for oral cancer images. In this procedure, after preprocessing and segmenting the area of interest of the inputted images, the useful characteristics are achieved. Then, some number of useful features are selected, and the others are removed to simplify the method complexity. Finally, the selected features move into a support vector machine (SVM) to classify the images by selected characteristics. The feature selection and classification steps are optimized by an amended version of the competitive search optimizer. The technique is finally implemented on the Oral Cancer (Lips and Tongue) images (OCI) dataset, and its achievements are confirmed by the comparison of it with some other latest techniques, which are weight balancing, a support vector machine, a gray-level co-occurrence matrix (GLCM), the deep method, transfer learning, mobile microscopy, and quadratic discriminant analysis. The simulation results were authenticated by four indicators and indicated the suggested method’s efficiency in relation to the others in diagnosing the oral cancer cases.

## 1. Introduction

One of the most common cancers is oral cancer, and many factors are involved in the development of this cancer. These proliferating cells accumulate together and form cancerous masses, which sometimes invade other parts of the body and cause problems that can be dangerous to people. Oral cancer is one of these cancers that can be dangerous for people if it is not diagnosed in time. Cancers of the cheeks, lips, floor of the mouth, tongue, sinuses, hard and soft palate, and lungs (throat) are types of this cancer that will be deadly if not detected and cured in the beginning stages [1]. Oral cancer can appear on the lips or other parts of the mouth with the tissues inside the gums, lips, and tongue. Oral cancer often changes parts of the skin [2]. For example, it causes the growth of thick tissue or the appearance of wounds that do not heal, even over time.

This is a kind of cancer of the neck and head and falls into the category of cancers of the mouth and throat. Based on a 2022 report of the American Cancer Society, it is considered to account for almost 3% of cancers diagnosed in the USA, meaning approximately 54,000 different records of oropharyngeal cancer or oral cavity cancer and around 11,230 deaths [3].

This has been recognized as the sixth-most common cancer at a world level, and its diagnosis and treatment are the responsibility of oral and ENT specialists [4]. Today, cancer treatment has become one of the most important challenges in medical society. Despite advances in medical science in recent years, researchers have yet to find a definitive cure for cancer. It is difficult to diagnose the symptoms of oral cancer, especially if it occurs in the throat. Sometimes, doctors do not even notice the important signs of oral cancer. To diagnose oral cancer, it is best to see a doctor who specializes in this field so that they can check the symptoms by performing a series of special tests and diagnose oral cancer in a timely manner [5]. Recently, artificial intelligence-based systems have been turned into a useful auxiliary for this purpose. They diagnose and treat various types of cancer, including oral cancer. If not detected and cured in the beginning stages, it will be life-threatening and cause problems for people, as mentioned before.

Correct cancer diagnosis has become one of the most important challenges facing medical societies today. Despite advances in medical science in recent years, researchers have not yet found a definitive system for cancer diagnosis [1,6]. To diagnose oral cancer, it is better to be checked by a doctor who specializes in this field so that they can check the symptoms by performing a series of special tests and recognizing them. However, because of the repetitive works that have been conducted by the specialists, the probability of errors has been increased. Different research works have been conducted based on artificial intelligence for this purpose.

Sharma et al. [7] applied a Possible Neural Network (PNN) and General Regression (GR) to diagnose and treat oral cancer at an early stage. According to the spread of cancer in India, the early detection of the disease can significantly help in the treatment of the illness [8]. In this study, the NN technique and GR approach are used to identify the disease. In this research, 35 traits and 1025 records are used to diagnose this disease. The results of the designed procedure showed that the accuracy of the suggested approach is 80%. Therefore, the capability of the PNN-GR design to recognize the disease was satisfactory. Finally, they developed this technique to identify the disease.

Bhandari et al. [9] extracted malignant lesions of the oral squamous tissue using a Deep Neural Network (DNN) to increase the efficiency of illness detection. In this research, they used the artificial intelligence technique for the easy calculation of the problem. The proposed technique was a deep learning algorithm (DLA). They also used the loss functions technique and the sigmoid function to reduce the error of the DLA. In the present study, datasets of four types of oral cancer have been used. The proposed solution showed that it could detect the disease with 96.5% accuracy [9]. Also, the proposed technique reduced the calculation time by about 35 milliseconds. This method can provide more accurate classification by reducing the gap in the data training stage. Therefore, the technique presented with these advantages could have the best identification for identifying the illness.

Hurvitz et al. [10] utilized a computational approach to the analysis of salivary exoderms for diagnosing the disease. For this study, they used the saliva of 21 patients to diagnose the disease. In this study, a support vector machine (SVM) has been applied to categorize and analyze the disease. Finally, the sensitivity and characteristics of the technique in disease classification were evaluated. The model performance evaluation showed that the model validation accuracy was estimated at 89% [10]. The results of the proposed method also showed that the model with 100% sensitivity, 89% specificity and 95% accuracy can classify the disease.

Speight et al. [11] analyzed the approach of Artificial Intelligence (AI) in identifying and preventing cancer. The identification of oral diseases, especially oral cancer, helps dentists to follow up and treat the disease. For this objective, the presented technique was evaluated on 1662 samples. The results of evaluating the performance of the technique designed showed that artificial intelligence can detect and identify about 74% of the disease with a sensitivity of 99%, an accuracy of 80%, and a reliability of 0.99%.

Chen et al. [12] designed an oral assessment design by the Neural Network (NN). According to the low precision of the conventional oral language scoring technology, they used deep learning technology to increase performance. In this study, a short-term memory neural network (LSTM) and Convolutional Neural Network (CNN) were used to classify the symptoms of the illness. The achievements of evaluations indicated that the designed approach has the best scoring and the highest efficiency for diagnosing the disease.

As observed from the literature, different kinds of artificial intelligence (AI) methods are used for oral cancer diagnosis. However, metaheuristics indicate better results in this case. However, different configurations of the metaheuristics in different parts are used for the diagnosis of oral cancer, and each of them have their disadvantages, like local optimization and a low convergence speed.

The present study proposes a new pipeline method that utilizes rough set theory and an amended version of the competitive search optimizer for the efficient diagnosis of oral cancer. The main contributions of this research can be summarized as follows:The development of a novel pipeline method for the diagnosis of oral cancer: The proposed method integrates various stages of the diagnosis process, including preprocessing, image segmentation, feature extraction, feature selection, and classification, into a single pipeline system. This approach enables the efficient and accurate diagnosis of oral cancer, which can ultimately lead to better patient outcomes.The utilization of rough set theory and an amended version of the competitive search optimizer for feature selection and classification: The proposed method employs rough set theory and an amended version of the competitive search optimizer for optimizing the feature selection and classification steps in the diagnosis process. This approach enhances the efficiency and accuracy of the diagnosis system by selecting the most informative features and optimizing the classification algorithm.A comparison of the proposed method with state-of-the-art techniques: The proposed method is compared with several other state-of-the-art techniques, including weight balancing, a support vector machine, a gray-level co-occurrence matrix (GLCM), the deep method, transfer learning, mobile microscopy, and quadratic discriminant analysis. The comparison analysis demonstrates the superiority of the proposed method over other techniques in terms of accuracy and effectiveness in diagnosing oral cancer cases.The validation of the proposed method on the Oral Cancer (Lips and Tongue) images (OCI) dataset: The proposed method is validated on the Oral Cancer (Lips and Tongue) images (OCI) dataset, which is a well-established dataset in the field of oral cancer diagnosis. The validation results confirm the efficiency and accuracy of the proposed method in diagnosing oral cancer cases.

In general, the proposed method holds significant potential for improving the accuracy and efficiency of oral cancer diagnosis, which can ultimately lead to better patient outcomes. The contributions of this research are expected to have a significant impact on the healthcare industry and the field of computer-aided diagnosis systems.

## 2. Dataset Description

In this study, for analyzing the proposed method, the “Oral Cancer (Lips and Tongue) images (OCI) dataset” has been employed. The database is accessible from the Kaggle website [13]. The dataset includes a set of lips and tongue images classified in cancerous and non-cancerous collections. The images were captured in different ENT hospitals of Ahmedabad and categorized with the help of ENT clinicians. This dataset collected 87 sets of oral cancer images and 44 sets of oral non-cancer images to be used by the researchers for different medical imaging purposes. The format of all images is “*.jpg”. The dataset can be reachable from https://www.kaggle.com/shivam17299/oral-cancer-lips-and-tongue-images (accessed on 18 March 2023). Figure 1 displays some examples of the non-cancerous and cancerous cases from the OCI dataset.

## 3. Image Pre-Processing

Image preprocessing is a process for solving issues in taking images that occur at medical imaging, like light or noise, and for correcting them intelligently [14]. In this process, some disruptions may exist because of various field brightness levels, reception high-frequency, and issues with remote orientation that AI and image processing (IP) help to correct, and they are usually taken by default on all images.

### 3.1. Noise Cancellation

There are several ways to help improve an image noise removal; therefore, choosing the correct method has a significant impact on achieving the desired image, and each of the noise removal methods is considered a specific problem [15]. For example, the method used to remove satellite image noise may not be suitable for medical image noise removal.

Image noise is assumed to be an independent process or as part of the processing. In part of the processing case, image noise cancellation has been applied to enhance the precision of different IP optimizers, including recording or categorization [16]. In the other case, noise cancellation is used to improve the image quality for visual inspection, which is important to protecting the relevant image information. The main purpose of image noise cancellation is the recovery of the optimum initial image estimates from the noise version. Some noise cancellation techniques such as the stochastically connected random field model [17], autoencoder [18], and median [19] are presented.

In this study, spatial filtering has been used for this purpose. The median filter (MF) is a simplified nonlinear operator that substitutes the window’s median pixel with the average amount of its surroundings. MF’s sliding window is 7 × 7 [20]. This is especially efficient for removing pulses or sharp points of noise. This filter is one of the most widely used filters for medical imaging. The pseudocode of the median filter has been illustrated below:
Enter input image “*im*” with the size M×N.Form a (M+2)×(N+2) matrix with zero adding to the input image sides.Employ a 7×7 mask.Take the component mask on the first row and column ($im_{1,1})$.Select all component masks and calculate the median.Assume the mean value of the sorted array and replace $im_{1,1}$ with the median value.Change the mask to the following component.If the image components are replaced by the median value, go to (9); else, go to (4).End

Figure 2 shows a sample noise removal of a pepper and salt noisy image using MF.

As can be observed in Figure 2, after performing the median filtering, a satisfying improvement has been achieved.

### 3.2. Contrast Enhancement

The process of image contrast enhancement includes alterations in the pixel intensiveness of the inputted image; thus, the outputted image seems improved perceptually and intuitively. Consequently, the target of image modification is to enhance the perception or interpretation of the information included in it for the looker or to present a better input for the automation system [6]. Contrast is a main quality element for processing various images. The information of the image disappears in regions that are evenly and too focused if an image contrast is too focused, too bright, or too dark. Thus, the image contrast must be improved to indicate the complete information of the inputted image. Based on image contrast enhancement, in some cases, even after the image pre-processing stage, it might not include the proper quality or adequate resolution to show the images’ information [21]. Therefore, images might need specific adaptations regarding scatter and brightness. IP and AI have been applied in medical imaging for this purpose [22]. Adjusting the adaptive histogram equalization (AHE) is the most common method of contrast enhancing, which has been applied in about all image kinds because of its comparatively proper performance and simplicity. The basic GHE concept is to reprogram the gray surfaces of the inputted image by the unvarying expansion of its probability density function (PDF).

AHE levels the image histogram’s dynamic range, enhancing the contrast of the inputted image fully. It cannot be proper for implementation in medical imaging systems due to changes in the average image brightness [23]. This technique shows a tendency to produce a problem of intensity saturation, make disturbing artificial effects, and increase noise in the outputted image due to over-enhancement [24]. In other words, conventional AHE tends to oversimplify contrast in near-constant regions due to the fact that the histogram is so focused in those regions [15]. Consequently, it might increase noise in near-fixed regions. Contrast-limited adaptive histogram equalization (CLAHE) is a type of AHE where the enhancement of contrast is restricted so that the noise enhancement issue is reduced.

In this process, the enhancement of contrast is converted in the neighborhood of a certain pixel amount with a function slope. This is according to the neighborhood slope, which is the cumulative distribution function (CDF) and, hence, the histogram value in that pixel’s value. Before calculating CDF, CLAHE reduces amplification by lowering the histogram to a predefined amount. This slope limits the CDF and thus the transformation performance. The amount when the histogram is cut, the pretended clip size, is dependent on the histogram normalization and, thus, the neighborhood size. Usual amounts restrict the achieved intensification from three to four. Leaving aside the part of the histogram that is too clamped, redistributing it evenly between the histogram buckets is beneficial. The pseudocode of the CLAHE is given below (Algorithm 1):
**Algorithm 1** CLAHE                    Input: Initial Image I;
Resizing I to M × M; Decompose I → (n) tiles; (n)$\leftarrow \;\frac{M\times M}{m\times m}$;$H_{n}\leftarrow$ histogram(n); // histogram of a m×m tile;Clip limit: $C_{L}\leftarrow N_{cl}\times N_{avg};$
                    //$N_{avg}\leftarrow \frac{N_{x}\times N_{y}}{N_{gray}}$;                    //$N_{avg}\to$ gray levels number in the tile;                    //$N_{x}$, $N_{y}\to$ pixels number in the x, y tile dimensions;                    //$N_{cl}\leftarrow$0.002// normalized contrast limit;
4.Clipping of $H_{n}$ using $C_{L}$;
                    //For gray levels higher than $C_{L}$; let $N_{\sum cl}$ pixels be clipped;
5.$\frac{N_{\sum cl}}{N_{gray}}\to N_{cp}$ pixels → distribution over the remaining pixels;
                    //contrast limited histogram of each tile after pixel distribution;
6.CLAHE(n) ← Equalization of contrast-limited tile histogram by (1);7.$I_{c}\leftarrow$ bilinear interpolation of CLAHE-processed n tiles;//integrating surrounding tiles                     Output: CLAHE-processed image $I_{c};$


Figure 3 shows a sample contrast enhancement for a cancerous oral image based on the GHE method.

## 4. Image Segmentation

Image segmentation is one of the image processing operations that causes the separation of different parts of the image and, in fact, separates the lesion from the background [25]. Image segmentation divides the image into meaningful pixel groups for the ease of image analysis and processing and corrects distorted borders. Here, an improved version of the K-means algorithm based on Rough Set Theory (RST) has been used. The RST can be considered as a tool for discovering data dependencies and reducing attributes in a dataset, using only data and without the need for additional information. The RST includes sets with fuzzy borders. The conception of the rough set theory contains a set associated with some information. The RST is a useful tool in data with uncertainty, which was first presented by Pawlak [26]. Based on the RST, a rough number includes a low-limit (L) boundary, a high-limit (U) boundary, and a rough border distance that relates to the original data. Consequently, there is no need for extra information for better understanding by the experts.

This assumes a set U with an arbitrary member, Y, and an R number of the class t which covers the U members. By considering the classes sequentially as $G_{1}<G_{2}<\cdots <G_{t}$, the lower, upper, and boundary regions for the G class have been achieved by the following equations:



(1)
$$\underline{G_{q}}=⋃\left(Y\in U|R\left(Y\right)\leq G_{q}\right)$$





(2)
$$\bar{G_{q}}=⋃\left(Y\in U|R\left(Y\right)\geq G_{q}\right)$$





(3)
$$G_{\left[q\right]}=⋃\left(Y\in U|R\left(Y\right)\neq G_{q}\right)=⋃\left(Y\in U|R\left(Y\right)>G_{q}\right)\cup ⋃\left(Y\in U|R\left(Y\right)<G_{q}\right)$$



Based on the above equations, the rough numbers in the lower and upper boundaries are given below:



(4)
$$\underline{lim}\left(G_{q}\right)=\frac{1}{M_{L}}\sum R\left(Y\right)|Y\in \underline{G_{q}}$$





(5)
$$\bar{lim}\left(G_{q}\right)=\frac{1}{M_{L}}\sum R\left(Y\right)|Y\in \bar{G_{q}}$$





(6)
$$RN\left(G_{q}\right)=\left\lceil \underline{lim}\left(G_{q}\right),\bar{lim}\left(G_{q}\right)\right\rceil$$



It can be observed that the difference between the lower and upper boundaries limits the rough boundary distance [26]. Likewise, the rough boundary distance indicates the uncertainty such that the upper value shows higher uncertainty.

K-means is the main idea for the segmentation of the images. This algorithm is an unsupervised learning technique applied for solving clustering issues in data science or machine learning. Unsupervised learning is a case in which, according to the data we have, the correct solution is not known, and these data have the same label or no label at all. Then, a dataset is provided to the algorithm that does not have a specific structure; the unsupervised learning algorithm (there are different types, such as a K-means algorithm, hierarchical clustering algorithm, etc.) determines what data should be placed in a cluster.

A technique of vector quantification is K-means clustering, which is initially taken from signal processing and is well known for data mining assessment. The purpose of this method is to observe K into n clusters, where observations relate to a cluster with the closest average to it, this mean being used as an example. This works by the following phases [27]:
      Step 1: To decide on the clusters’ number, the number K is selected.      Step 2: Initialize random cluster centers ($\mu_{1},\mu_{2},\ldots \mu_{k}\in R^{n}$)      Step 3: K points are selected randomly or by calculation. (This can be something other than the input dataset.) Based on the following code, the Euclidean distance is used to select the centers.      For every i, set $c^{i}=\underset{i}{\min}{\left\|x^{i}-\mu_{j}\right\|}^{2}$ [27].      For each j, set $\mu_{j}=\frac{\sum_{i=1}^{m}l\left(c^{i}=j\right)x^{i}}{\sum_{i=1}^{m}l\left(c^{i}=j\right)}$      Step 4: Compute the average and locate a new center for clusters.      Step 5: The third step is repeated, meaning that each database is assigned to the newest and nearest center of clusters.      Step 6: If a change happens again, phase four is performed again, and the algorithm ends.      Step 7: The model is ready.

Based on the conception of the rough set and K-means algorithm, the rough K-means algorithm, as a useful tool, is used. According to grey-scale image segmentation, the main issue is to segment clusters between blurred boundaries. Using the RST, an image has been specified, including lower and upper approximations. The rough K-means model has been achieved by the following equation:



(7)
$$C_{j}=\left\{\begin{array}{l} w_{lower}\times \frac{\sum_{v\in A}v_{j}}{\left|\underline{A}(x)\right|}+w_{upper}\times \frac{\sum_{v\in (\bar{A}\left(x\right)-\underline{A}(x))}v_{j}}{\left|\bar{A}(x)-\underline{A}(x)\right|},\;if\;\bar{A}\left(x\right)-\underline{A}\left(x\right)\neq \varphi \\ w_{lower}\times \frac{\sum_{v\in A}v_{j}}{\left|\underline{A}(x)\right|}+,\;\;\;otherwise \end{array}\right.$$



Here, by performing 4 × 4 windowed GLRM features and separating them into lower or upper approximation, the rough K-means model is achieved by the following:
-4 × 4 windowed GLRM features have been considered as a lower approximation member of A(x).-In the event that the GLRM features are a portion of the lower approximation A(x), then, it is similarly a portion of the upper approximation A(x).-In the event that the GLRM features do not depend on lower approximations, A(x), they relate to two or more upper approximations A(x).


To implement rough set theory into the K-means method, the following conception has been considered:
(8)$$T=\left[j:d\left(v,c_{j}\right)\leq Threshold\;and\;i\neq j\right]$$
(9)$$d\left(v,c_{i}\right)=\underset{1\leq j\leq k}{\min}d\left(v,c_{j}\right)$$
where $C_{i}$ defines the clusters center, v is the GLRM features, and $d\left(v,c_{j}\right)$ specifies the distance between the windowed GLRM features.
Specify N random clusters.Specify $w_{lower}$ and the threshold value.For all clusters, define T and $d\left(v,c_{j}\right)$ using Equations (8) and (9).Segment GLRM features approximations based on the criteria.Calculate a new center for the cluster.If the termination criteria have been reached, go to (7); else, go to (3).End

## 5. Features Extraction

Features are recognizable structures in an image that are extracted from the input image by measurements. Feature extraction (FE) is a significant step in machine learning. Extracting desirable features improves classification accuracy. Features extracted by mathematical relations can be expressed by mathematical equations [28]. Features fall into two groups of local and global features. The description of the features is considered in three categories of color, shape and texture. A color feature (such as a color histogram) is used in image retrieval [29,30,31]. The color histogram has been considered because of its easy, stable, and effective implementation and low computations. FE techniques are broadly applied in images classification [32]. An image’s features contain all identifiable frameworks that have been achieved based on the image’s nature [33,34,35,36]. The database of images can include color and gray images. It can also be applied in various fields such as handwriting recognition, face recognition, signature verification, and cancer scanning. Here, texture, geometric, and statistical features have been used. Texture features are used for analyzing entropy, contrast, correlation, energy, and homogeneity. Statistical features contain the standard deviation, variance, invariant moments, mean, and geometric features, for analyzing the area, solidity, equivalent diameter, size, eccentricity, perimeter, irregularity index, and convex area. Table 1 illustrates the utilized features for extraction.

where a and b define, respectively, the minor and major axes, $\left(i,j\right)$ defines the intensity amount of the point (i,j), MN specifies the image size, $b_{p}$ describes the exterior side length of the boundary pixel p, and μ and σ describe the mean value and the standard deviation (STD) of the pixels, respectively.

To simplify the process, we need to reduce the utilized features by removing some useless cases and keeping useful features [38]. The method of features selection is given in the following.

Features selection makes choosing the model feature easier. This reduces the cost of calculations. By removing useless features, the model becomes clearer and more comprehensive. It also speeds up the learning process, reduces storage space, and improves performance such as accuracy. As a result, feature selection algorithms are essential to reducing data dimensions in high-dimensional data.

In this study, a minimization function is defined for solving the feature selection problem. This function has been defined below:
(10)$$C_{i}=w_{1}\times {ACC}_{i}+w_{2}\times \frac{\sum_{i=1}^{M_{b}}F\left(FS\left(i\right)\right)}{\sum_{j=1}^{N_{b}}F\left(j\right)}$$
where $w_{1}+w_{2}=1$.

As can be observed, the above function is designed based on an F-score and classification accuracy (${ACC}_{i}$).

Here, the accuracy is considered in a higher priority; therefore, $w_{1}=0.6$ and $w_{2}=0.4$. The value of weights is achieved after some trials and errors. The accuracy classification rate is obtained by the following formula:
(11)$${ACC}_{i}=\frac{\sum_{i=1}^{\left|N_{TS}\right|}AS\left(l_{i}\right)}{\left|N_{TS}\right|},\;\;\;l_{i}\in N_{TS}$$
where
(12)$$AS\left(k\right)=\left\{\begin{array}{l} 1,\;\;\;if\;classify\;\left(k\right)=ec\\ 0,\;\;\;otherwise \end{array}\right.$$

And $N_{TS}$ and ec describe the test set numbers and class of element l, respectively.

Another real-valued set is the F1-score. Considering the training vectors $X_{k|k=1,2,\ldots ,l}$, the F1-score of the $i^{th}$ feature has been obtained as follows:
(13)$$F\left(i\right)=\frac{{\left({\bar{x}}_{i}^{+}-{\bar{x}}_{i}\right)}^{2}+{\left({\bar{z}}_{i}^{-}-{\bar{z}}_{i}\right)}^{2}}{\frac{1}{l^{+}-1}\sum_{j=1}^{l^{+}}{\left(x_{j,i}^{+}-{\bar{x}}_{i}^{+}\right)}^{2}+\frac{1}{l-1}\sum_{j=1}^{l^{-}}{\left(l_{j,i}^{-}-{\bar{l}}_{i}^{-}\right)}^{2}}$$
where $\bar{x}$ is the $i^{th}$ average amount of the feature, $l^{+}$ and $l^{-}$ specify the positive and negative instance numbers, respectively, ${\bar{x}}_{j,i}^{+}$ and ${\bar{x}}_{j,i}^{-}$ define the $i^{th}$ feature of the $j^{th}$ positive and negative instances, respectively. In Equation (13), F(i) defines a function to find the score of the current features and is given below:
(14)$$F\left(FS\left(i\right)\right)=\left\{\begin{array}{l} instance\;i,\;if\;i\;is\;selected\\ 0,\;\;\;if\;i\;is\;not\;selected \end{array}\right.$$

By considering the F1-score for the $i^{th}$ feature, we have:
(15)$$F_{i}=\frac{\sum_{i=1}^{l}{\left(x_{j}^{i}-{\bar{x}}_{j}\right)}^{2}}{\sum_{i=1}^{l}\frac{1}{l_{i}-1}\sum_{k=1}^{l_{i}}{\left({\bar{x}}_{k,j}^{i}-{\bar{x}}_{j}^{i}\right)}^{2}}$$
where, ${\bar{x}}_{k,j}^{i}$, ${\bar{x}}_{j}$, and ${\bar{x}}_{j}^{i}$ represent the $j^{th}$ feature of the instance, the mean value of the $j^{th}$ feature for the total database, and the $i^{th}$ database, respectively.

So, to obtain a feature selection, Equation (10) should be minimized. This study uses an amended version of the competitive search algorithm for this purpose.

## 6. Amended Competitive Search Algorithm

### 6.1. The Competitive Search Optimizer (CSO)

After introducing the intellectual source of the CSO, the main framework and mathematical model of this algorithm are presented. The principle of CSO optimization has also been analyzed.

i.Source of thought

The competitive search algorithm was created using human social activities and differs in this respect from other algorithms derived from the physical laws and characteristics of animals. Different competitive programs, such as Pop Idol and America’s Got Talent, those shown on television, follow almost the same process. In these competitions, participants take a learning course after ranking from various aspects to prepare for the next stage. At the end, the participants are evaluated, and the best one is selected, which is an optimization process.

First, we assumed that a competition program has different competition scoring standards, containing dancing, singing, height, weight, and appearance. All participants in the competition are evaluated by a comprehensive test and ranked according to their scores. Two groups are formed based on the specified ranks. These two groups are excellent and general groups that are trained to prepare for the next stage of the competition through different methods. Finally, after learning and evaluating successively, the program hero is selected.

ii.Framework of the algorithm and mathematical modeling

Due to the fact that different competitions have different rules, the rules of the competition are presented, and its mathematical model is made.

Rule 1: Participants are evaluated according to some standards, and each participant is given points; then, two excellent and general groups are formed according to the participants’ points.

Rule 2: Participants have various capabilities in learning. Over time, in the game, the ability to learn changes randomly. Each group specifies a learning capability threshold, and anything higher than this amount is considered as a powerful learning capability. Also, the lower is considered as a normal learning capacity.

Rule 3: After all participants have completed each course, the strong learner has a more various range of learning than the average learner in the excellent group: The stronger group (the first group based on ranking) has a larger learning range, so the learning group of the second group is relatively smaller.

Rule 4: In the general group, participants’ learning is based on their abilities in such a way that those who are more capable of learning focus more on improving themselves. But those who have the ability to learn normally expect to fail through themselves more.

Rule 5: If a participant’s ability to learn is greater than a certain value, it is considered as a reference behavior. And participants learn from the best participant indicators based on their abilities.

Rule 6: Some participants withdraw from the competition at the end of each round for various reasons and are replaced by new participants such that the number of participants in each round does not change. The main evaluation indicators and the ability of new participants are random.

In the competition simulation, the virtual contestants are embraced for the competition. The number of participants is given in the following formula [39]:
(16)$$Y=\left[\begin{array}{c} \begin{array}{ccc} Y_{1,1} & Y_{1,2} & \begin{array}{cc} \ldots & Y_{1,d} \end{array} \end{array}\\ \begin{array}{ccc} Y_{2,1} & Y_{2,2} & \begin{array}{cc} \ldots & Y_{2,d} \end{array} \end{array}\\ \begin{array}{c} \begin{array}{ccc} \vdots & \vdots & \begin{array}{cc} \vdots & \vdots \end{array} \end{array}\\ \begin{array}{ccc} Y_{n,1} & Y_{n,2} & \begin{array}{cc} \ldots & Y_{n,d} \end{array} \end{array} \end{array} \end{array}\right]$$
where various indicators evaluated for contest participants are indicated by d; in other words, it states the problem later. Equation (2) presents the fitness value of each participant:
(17)$$F_{y}=\left[\begin{array}{c} f(\left[y_{1,1,},y_{1,2},\ldots y_{1,d}\right])\\ f(\left[y_{2,1,},y_{2,2},\ldots y_{2,d}\right])\\ \begin{array}{c} \vdots \\ \begin{array}{c} \vdots \\ f(\left[y_{n,1,},y_{n,2},\ldots y_{n,d}\right]) \end{array} \end{array} \end{array}\right]$$
where the number of contestants is shown by n, and the value in all rows describes the fitness value attained by each contestant.

In the competitive search algorithm, after evaluating all participants, their fitness values are ranked after each round of the competition. Two groups of participants are formed based on the fitness value: excellent and general [40,41]. In the excellent group, the contestants with top rankings and more powerful learning capability due to the restriction of their upward space will have less progress than the contestants with more powerful learning abilities but lower rankings. Participants will progress more in higher rankings with a stronger learning ability [42]. Equation (3) states the update of the parameters of all indexes of the excellent with a more powerful learning capability:



(18)
$$Y_{i,j}^{t+1}=Y_{i,j}^{t}+A\left(i\right)*S_{1}*\rho *\left(u_{b}^{j}-l_{b}^{j}\right)\;if\;A(i)>L_{1}\;S_{1}=\left(U_{B}*rand\left(1\right)+L_{B}\right)\%$$



Equation (4) indicates updating the index parameters of excellent contestants with the top ranking and normal learning capability:



(19)
$$Y_{i,j}^{t+1}=Y_{i,j}^{t}+A\left(i\right)*S_{2}*\rho *\left(u_{b}^{j}-l_{b}^{j}\right)\;if\;A(i)\leq L_{1}\;S_{2}=\left(L_{B}*rand\left(1\right)\right)\%$$



The search range functions of strong learning capability and general learning capability are indicated by $S_{1}$ and $S_{2}$, and the present iteration number is indicated by t; j represents the number of dimensions that Y is placed in; j=1,2,3,4…d; the value of the $j^{th}$ evaluation index of the $i^{th}$ contestant is indicated by $Y_{i,j}$, and this expresses the location information in the $j^{th}$ dimension; constants are shown by $U_{B}$ and $L_{B}$; the minimum and higher limits of the function in the $j^{th}$ dimension search range are represented by $l_{b}^{j}$ and $u_{b}^{j}$; $A\left(i\right)$ represents the learning capability of the present participant; ρ indicates the amount randomly obtained from the matrix [−1, 0, 1]; the threshold amount illustrating the power of the learning capability in the excellent group is shown by $L_{1}$, which belongs to the matrix (0, 1).

From the formulas presented in (3) and (4), it can be concluded that the distinction in updating the position of the participants in the excellent group is only in $S_{1}$ and $S_{2}$. The search range for each dimension of participants who have a normal learning power is between 0%$\;and\;L_{B}$%, and this range is for participants with a strong learning power between $L_{B}\%\;and\;U_{B}\%$. This causes the search range to be more exhaustive. The learning direction of participants is represented by a random number with a value equal to [−1, 0, 1]. In other words, when this number is equal to −1, it means that the participants are learning in the opposite direction, and when it is 1, the participants are learning in a positive direction, and the number 0 indicates that the participants are not learning in this round [43]. According to Rule (4), in the general group, participants can study for each evaluation round. Formula (5) presents the updated performance of each index.

(20)$$Y_{i,j}^{t+1}=\left\{\begin{array}{c} Y_{i,j}^{t}+\alpha .Q.D\;\;\;\;\;\;if\;A(i)>L_{1}\\ Y_{i,j}^{t}.L_{2}.F.A\left(i\right)\;\;\;\;\;\;if\;A(i)\;\leq L_{1}\;\;\;\;\;\; \end{array}\right.\;F=P.o$$
where α indicates the random amount between −1 and 1, Q defines a random amount between 0 and 2, F indicates a negative agent, $L_{2}$ and D are the 1×d matrices, each of the components in the matrix D are 1, and the components in $L_{2}$ are randomly allocated with 1 and −1; o is a random element, when the positions of contestants are updated, this element randomly choose from the matrix [0.1, 0.2, 0.3, 0.4, 0.5]; P is a standard, normal distribution with a variance of 1 and a mean of 0.

As stated in Rule (5), reference behavior appears for a participant whose ability to learn exceeds a set amount: the participants learn from the best participant based on their learning ability. Equation (6) describes this process:
(21)$$Y_{i,j}^{t+1}=Y_{i,j}^{t+1}+\left(Gbest\;Y_{j}^{t}-Y_{i,j}^{t+1}\right)*A\left(i\right)\;\;\;\;\;\;if\;A(i)>L_{3}$$
where the $j^{th}$ index value of the best participant in the $i^{th}$ iteration is indicated by $Gbest\;Y_{j}^{t}$; $L_{3}$ indicates the reference threshold between (0, 1); $\left(Gbest\;Y_{j}^{t}-Y_{i,j}^{t+1}\right)$ describes the split between the present participant and the optimal one. The present participant can move closer to the best participant by multiplying $\left(Gbest\;Y_{j}^{t}-Y_{i,j}^{t+1}\right)$ by the learning ability A(i).

Updating and training the evaluation indicators of all participants are accomplished by (3)−(6). As stated in Rule (6), after each round of the competition, some participants cannot advance to the next stage, and in order to keep the number of participants constant, a corresponding number of participants are added randomly, and all learning abilities and indicator evaluations occur randomly. The pseudo-code of the competitive search optimizer is presented in Table 1, and the framework and basic phases of the algorithm have been stated (Algorithm 2).
**Algorithm 2** Competitive search algorithm framework      Procedure CSO (number of contestants n, maximum iteration G, number of excellent contestants EC,      number of contestants who withdrew after each round RC, $L_{1}$, and $L_{3}$)      The various indicators of n contestants are initialized, and the relevant parameters are defined:      1: $A=rand(\left[1\right],n)$      2: while(t < G)      3: Calculate the fitness value of each contestant and rank      4: for i=1:EC      5: Use (3) and (4) to update the indicators of the contestants      6: end for      7: for i=EC=1:n
      8: The (5) is used to update the indicators of the contestants      9: end for      10: for i=1:n      11: The (6) is used to update the indicators of the contestants      12: end for      13: Randomly eliminate RC contestants      14: Obtain updated indicators of contestants      15: t=t+1      16: end while

### 6.2. Amended Competitive Search Algorithm

While the original competitive search algorithm is a new, efficient optimizer for optimizing the problems of optimization, it may face some issues like the incorrect random replacement of the worst candidate or even the lack of a proper exploitation, which provides premature convergences [44,45,46]. Here, we consider a modification for improving the efficiency of the algorithm. This study uses the sine–cosine mechanism as chaos theory and opposite-based learning (OBL) to obtain a higher efficacy [47].

At first, the candidates that describe the worst cost of the epoch are selected to be updated, and the new position has been calculated as follows:



(22)
$$X_{worst}^{i}=\left\{\begin{array}{c} X_{worst}^{i}+a_{1}\times \sin\left(a_{2}\right)\times \left|a_{3}\times X_{best}^{i}-X_{worst}^{i}\right|\;\;\;\;\;\;a_{4}<0.5\\ X_{worst}^{i}+a_{1}\times \cos\left(a_{2}\right)\times \left|a_{3}\times X_{best}^{i}-X_{worst}^{i}\right|\;\;\;\;\;\;a_{2}\geq 0.5 \end{array}\right\}$$



Here, $a_{1}$, $a_{2}$, $a_{3}$, and $a_{4}$ represent some coefficients which are obtained as follows:
(23)$$a_{1}=\alpha -iter_{curr}\times \left(\gamma /iter_{max}\right)$$
(24)$$a_{2}=2\pi \times rand$$
(25)$$a_{3}=2\times rand$$
(26)$$a_{4}=rand$$
where γ describes a constant and $iter_{curr}$ and $iter_{max}$ describe the present and the maximum iterations, respectively.

The second modification is to use the OBL approach. If the rand value has less than the Jump Rate (JR) as a constant value, the updated candidates cost is promptly calculated, and their new members of the matching opposite are obtained; then, the best candidates are chosen as the best candidates, and then the cost assessment is applied by the candidates. Based on the concept of the competitive search algorithm, the updated positions, and the opposite-based learning policy, the performance of the competitive search algorithm is enhanced, and the drawbacks have been improved.

### 6.3. Algorithm Assessment

After designing the proposed amended competitive search algorithm, it is better to analyze the method’s efficiency. To evaluate the performance of the suggested technique, it was confirmed by four test functions, including Sphere, Rosenbrock, Ackley, and Rastrigin, and the results were put in comparison with several of the latest optimizers, which are the Supply–Demand-Based Optimizer (SDO) [48], Biogeography-Based Optimization (BBO) algorithm [49], and Emperor penguin optimizer (EPO) [50]. Table 1 illustrates the parameter value of the studied optimizers (Table 2).

The coding for optimizers is performed by a MATLAB R20190 environment on a PC with Intel Core i7-6700, a 3.40 GHz CPU, and 16 GB of RAM. Table 3 indicates the applied test functions to validate.

The population number and the highest iteration for the algorithms were, respectively, considered 60 and 250. The algorithms validation has been conducted 35 times independently to achieve a fair comparison based on the standard deviation results of the solutions. For analyzing the efficiency of the compared methods, they studied based on their average value and standard deviation values. The results of the comparison of the suggested amended competitive search optimizer with the investigated optimizers are reported in Table 4.

As can be observed, for the Sphere function, the ACSO algorithm achieved the lowest average value of 109.542, which is significantly lower than the average values of WOA (562.128), HHO (435.876), and FOA (364.529). Moreover, the standard deviation of the ACSO algorithm (96.637) is much lower than the standard deviations of HHO (201.563) and WOA (245.154), indicating that the ACSO algorithm is more stable and consistent in its performance. For the Rastrigin function, the ACSO algorithm also achieved the lowest average value of 1.3647 × 10^−5^, which is significantly lower than the average values of WOA (232.169), HHO (145.364), and FOA (73.0101). Additionally, the standard deviation of the ACSO algorithm (0.038 × 10^−5^) is significantly lower than the standard deviations of HHO (81.824) and WOA (94.588), indicating that the ACSO algorithm is more stable and consistent in its performance. For the Ackley function, the ACSO algorithm achieved a comparable average value of 2.097 × 10^−6^ in relation to the other algorithms, indicating that it is not significantly better or worse than the other algorithms. However, the standard deviation of the ACSO algorithm (1.052 × 10^−6^) is lower than the standard deviation of WOA (56.642), indicating that the ACSO algorithm is more stable and consistent in its performance. For the Rosenbrock function, the ACSO algorithm achieved an average value of 0.951 × 10^−2^, which is slightly worse than the average value of FOA (3.261). However, the standard deviation of the ACSO algorithm (0.121 × 10^−2^) is lower than the standard deviation of FOA (2.041), indicating that the ACSO algorithm is more stable and consistent in its performance. Overall, the numerical results presented in Table 3 demonstrate that the ACSO algorithm outperforms the other investigated optimization algorithms for the Sphere and Rastrigin benchmark functions and is comparable in performance for the Ackley benchmark function. For the Rosenbrock benchmark function, the ACSO algorithm performs slightly worse than the FOA algorithm in terms of the average value but is more stable and consistent in its performance. According to the results, using an amended competitive search algorithm with a lower average value delivers the highest accuracy among the other methods. This higher accuracy shows more validation of the suggested technique with the desirable values. Similarly, the lowest amount of the STD indicates the higher reliability of the proposed algorithm compared to the others. This technique has been used as an optimization solver in the feature selection process.

## 7. Data Classification

Via former labeled data, a design for predicting new data labels can be created, which is called classification [54]. This is an original sub-branch of data mining and machine learning and is defined by data gathered from previous practices. To obtain a proper categorization design, it is required to know the data and their configuration, besides the classes’ numbers (label–class–class). Even though it can sometimes be impossible to get acquainted with the type and structure of data, by simple familiarity, it is sometimes feasible to select the true categorization design. The two main types of classification are supervised and unsupervised approaches. In this study, we use supervised methodology. By labeled examples, the supervised methodology is able to use previously learned things for predicting coming events for new data. In analyzing a defined data group process, the optimizer creates a function to predict the output values. This will provide the aims of new entries after sufficient training. The output of the algorithm can be compared with the intended correct output, and its error can be found to change the design. Different kinds of classification are defined for this purpose. Herein, a modified SVM is used.

SVM contains a group of data with d dimensions, which is utilized for indicating the boundaries of the classes and for categorizing them. The best results for the SVM are achieved by the criterion for placing support vectors.

The major target of the support vector machine is to obtain the optimum data boundary as far as feasible from all groups and not to be responsive to other data points. The support vector machine aims to perceive the best surface for the decision by the following equation:
(27)$$z=\mathrm{sgn}\left(\sum_{i=1}^{M}y_{i}\alpha_{i}K(x,x_{i})+\beta \right)$$
where M specifies the training set number, y describes the class label between −1 and 1, x describes the dimensional test set, $x_{i}$ defines the $i^{th}$ training set vector, $K(x,x_{i})$ represents a kernel function, and $\alpha =[\alpha_{1}\ldots \alpha_{N}]$ and b define the model parameters. The selected features in this process are then injected into the classifier. To deliver optimal classification, the proposed amended competitive search algorithm is employed. This study presents the optimalization based on the optimal weights measurement of the SVM, which is carried out through lessening the mean square error (MSE) function. The mathematical equation of the MSE is given below:
(28)$$MSE=\frac{1}{2}\times \sum_{i}\sum_{j}{\left(z_{j}^{*}-z_{j}\right)}^{2}$$
Here, i and j represent the training samples’ number and the number of nodes in the output, and $z_{j}^{*}$ and $z_{j}$, respectively, describe the actual and the favored output. Afterward, the classifier is used for classifying the images into cancerous and non-cancerous oral cases.

## 8. Simulation Results

This paper designs an optimum pipeline method for detecting oral cancer. The procedure starts with a preprocessing stage for noise cancelation and contrast enhancement of the input images. The preprocessed images are then injected into an image segmentation system to segment the area of interest. Then, a feature extraction methodology is performed to achieve the segmented images’ features. Figure 4 shows the workflow of the entire system.

To simplify the method complexity, a feature selection based on the classification accuracy and F-score has been employed. Finally, an SVM was applied to classify the images based on selected features. As mentioned, the feature selection step and the SVM are optimized by an amended design of the CSO.

Here, to validate the accuracy of the suggested oral cancer diagnosis system, some different measurement indicators including the precision, recall, accuracy, and F-1 score have been used. In the following, the mathematical formulation of these indicators is given.
(29)$$Precision=\frac{1}{l}\times \sum_{i=1}^{l}\left(TP_{i}\right)\times \left(TP_{i}\times FP_{i}\right)$$
(30)$$Recall=\frac{1}{l}\times \sum_{i=1}^{l}\left(TP_{i}\right)\times \left(TP_{i}\times FN_{i}\right)$$
(31)$$Accuracy=\frac{\sum_{i=1}^{l}\left(TP_{i}+TN_{i}\right)}{TP_{i}+TN_{i}+FP_{i}+FN_{i}}$$
(32)$$F1=\frac{2\times Precision\times Recall}{Precision+Recall}$$
where TP, TN, FN, and FP signify the true positive, true negative, false negative, and false positive.

In the above equations, precision describes the correctness validation of the methods. Recall defines the capability of the classifier to obtain positive pixels, which indicates the total percentage value of the relevant instances that are retrieved correctly. The term accuracy indicates the proficiency examination of the diagnosis system, considering the relation values for correct approximation to the total estimations number. The F1 score, as the last term, can be achieved by the precision and recall.

The suggested oral cancer diagnosis system is applied to the Oral Cancer (Lips and Tongue) images (OCI) dataset, and the achievements have been put in comparison with some various diagnosis systems, including a gray-level co-occurrence matrix (GLCM) [55], weight balancing [56], a support vector machine [57], the deep method [58], transfer learning [59], mobile microscopy [60], and quadratic discriminant analysis [61]. Table 4 indicates the comparison assessment of the suggested technique against the others for oral cancer diagnosis.

According to Table 4 and Table 5, the proposed oral cancer diagnosis system outperformed all other techniques in terms of accuracy, precision, recall, and F1-score, achieving an accuracy rate of 94.65%. The second-best performing technique was the support vector machine (SVM), with an accuracy rate of 82.44%. The gray-level co-occurrence matrix (GLCM) technique and the deep method achieved the same accuracy rate of 82.44%. The transfer learning technique achieved an accuracy rate of 81.67%, while the mobile microscopy technique achieved an accuracy rate of 78.62%. The weight balancing technique and the quadratic discriminant analysis technique achieved accuracy rates of 78.62% and 74.81%, respectively. These results suggest that the proposed technique has significant potential for improving the accuracy and effectiveness of oral cancer diagnosis, which can ultimately lead to better patient outcomes. The high accuracy rate achieved by the proposed technique indicates its ability to distinguish between cancerous and non-cancerous tissues accurately, which is crucial for the early detection and timely treatment of oral cancer. Overall, the results of the comparison assessment demonstrate the superiority of the proposed technique over other state-of-the-art techniques for oral cancer diagnosis.

## 9. Conclusions

The study emphasized the importance of an efficient and accurate diagnosis system for oral cancer, which could lead to the early detection and prevention of potential fatalities. A novel pipeline approach was proposed which incorporated rough set theory and an amended version of the competitive search optimizer for optimizing the feature selection and classification steps in the diagnosis process. The proposed method was applied to the Oral Cancer (Lips and Tongue) images (OCI) dataset, and its efficiency was compared with several other state-of-the-art techniques, including weight balancing, a support vector machine, a gray-level co-occurrence matrix (GLCM), the deep method, transfer learning, mobile microscopy, and quadratic discriminant analysis. The proposed oral cancer diagnosis system demonstrated superior performance compared to the other techniques. It achieved an accuracy rate of 94.65%, surpassing all other methods. The second-best performing technique, the support vector machine (SVM), achieved an accuracy rate of 82.44%. The GLCM technique, the deep method, and the transfer learning technique achieved the same accuracy rate as SVM, while the mobile microscopy technique achieved an accuracy rate of 78%. Therefore, the simulation results indicated that the proposed method outperformed the other techniques in terms of effectiveness in diagnosing oral cancer cases. Therefore, the proposed method held significant potential for improving the accuracy and efficiency of oral cancer diagnosis, which could ultimately lead to better patient outcomes. The proposed method shows promising results for improving the accuracy and efficiency of oral cancer diagnosis. However, there are several areas for future research and limitations of this study that should be addressed. One area for future research is the evaluation of the proposed method on larger and more diverse datasets in order to validate its effectiveness in real-world scenarios. Additionally, while the proposed method outperforms state-of-the-art techniques in diagnosing oral cancer, it relies on feature engineering and may be prone to overfitting. Future research could explore alternative feature selection and extraction methods for mitigating these challenges. Another limitation of the proposed method is that it was evaluated on a single dataset. Further studies are needed to evaluate its performance on other datasets to assess its generalizability and robustness. Moreover, the proposed method’s effectiveness in clinical practice needs to be evaluated using real-world patient data. Despite these limitations, the proposed method has significant potential for improving the accuracy and efficiency of oral cancer diagnosis.

## Figures and Tables

**Figure 1 diagnostics-13-02454-f001:**
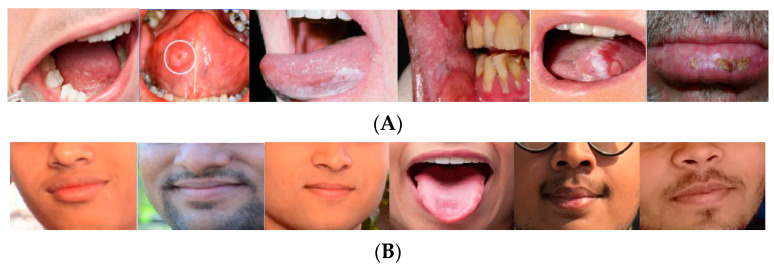
Some examples of the non-cancerous and cancerous cases from the OCI dataset: (**A**) cancer and (**B**) non-cancer cases.

**Figure 2 diagnostics-13-02454-f002:**
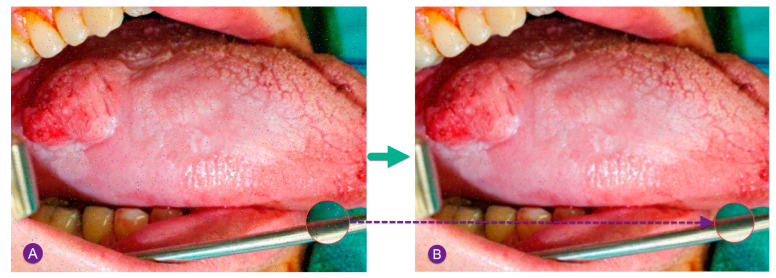
Sample noise removal of a pepper and salt noisy image using MF: (**A**) input noisy image by Gaussian noise, (**B**) image after median filtering.

**Figure 3 diagnostics-13-02454-f003:**
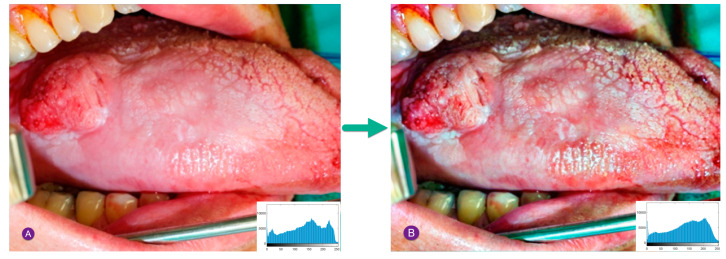
Sample contrast enhancement for a cancerous oral image by the GHE technique: (**A**) inputted image, (**B**) image after histogram improvement.

**Figure 4 diagnostics-13-02454-f004:**
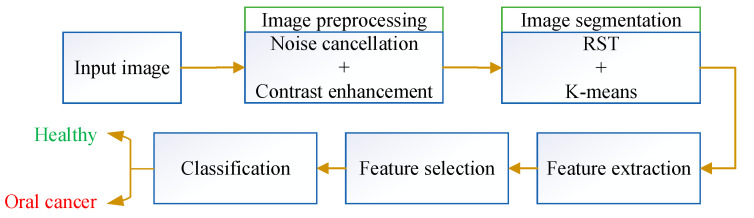
Workflow of the system.

**Table 1 diagnostics-13-02454-t001:** Features applied in this paper [37].

Feature	Formula	Feature	Formula
Correlation	$\sum_{i=1}^{M}\sum_{j=1}^{N}\frac{p(i,j)-\mu_{r}\mu_{c}}{\sigma_{r}\sigma_{c}}$	Rectangularity	$\frac{Area}{a\times b}$
Area	$\sum_{i=1}^{M}\sum_{j=1}^{N}p(i,j)$	Mean	$\frac{1}{MN}\sum_{i=1}^{M}\sum_{j=1}^{N}p(i,j)$
Solidity	Area/Convex Area	Entropy	$-\sum_{i=1}^{M}\sum_{j=1}^{N}p(i,j)\log p(i,j)$
Elongation	$\frac{2\sqrt{Area}}{a\sqrt{\pi }}$	Perimeter	$\sum_{i=1}^{M}\sum_{j=1}^{N}b_{p}(i,j)$
Homogeneity	$\sum_{i=1}^{M}\sum_{j=1}^{N}\frac{p(i,j)}{1+|i-j|}$	Variance	$\frac{1}{MN}\sum_{i=1}^{M}\sum_{j=1}^{N}\left(p(i,j)-\mu \right)$
Irregularity index	$4\pi \times Area/Perimeter^{2}$	Standard deviation	$\mathrm{var}iance^{\frac{1}{2}}$
Contrast	$\sum_{i=1}^{M}\sum_{j=1}^{N}p^{2}(i,j)$	Invariant moments	$\varphi_{1}=\eta_{20}+\eta_{02}$
Form factor	$F=\frac{Area}{a^{2}}$		$\varphi_{2}={\left(\eta_{20}-\eta_{02}\right)}^{2}+4\eta_{11}^{2}$
Energy	$\sum_{i=1}^{M}\sum_{j=1}^{N}p^{2}(i,j)$		$\varphi_{3}={\left(\eta_{30}-3\eta_{12}\right)}^{2}+{\left(3\eta_{21}-\mu_{03}\right)}^{2}$
Eccentricity	$2a^{-1}{\left(a^{2}-b^{2}\right)}^{0.5}$		$\varphi_{3}={\left(\eta_{30}+3\eta_{12}\right)}^{2}+{\left(3\eta_{21}+\mu_{03}\right)}^{2}$

**Table 2 diagnostics-13-02454-t002:** Parameter setting of the studied optimizers.

Algorithm	Parameter	Value
SDO [48]	MaxIteration	200
	MarketSize	40
	FunIndex	1
BBO [49]	Step size for the numerical integration of probabilities	1
	Immigration probability bounds per gene	[0, 1]
	Max immigration (I) and Max emigration (E)	1
	Habitat modification probability	1
	Mutation probability	0.005
EPO [50]	$\vec{A}$	[−1.5, 1.5]
	value of temperature ($T^{\prime }$)	[1, 1000]
	M	2
	f	[2, 3]
	S	[0, 1.5]
	l	[1.5, 2]

**Table 3 diagnostics-13-02454-t003:** Applied test functions for validation.

Type	Function Name	Function	Dim	Range	$F_{min}$
F1	Sphere	$F_{1}\left(x\right)=\sum_{i=1}^{n}x_{i}^{2}$	30	[−100, 100]	0
F2	Rosenbrock	$F_{3}\left(x\right)=\sum_{i=1}^{n-1}\left[100{\left(x_{i+1}-x_{i}^{2}\right)}^{2}+{\left(x_{i}-1\right)}^{2}\right]$	30	[−30, 30]	0
F3	Ackley	$F_{6}\left(x\right)=-20exp(-0.2\sqrt{\frac{1}{n}\sum_{i=1}^{n}x_{i}^{2}})-exp(\frac{1}{n}\sum_{i=1}^{n}cos(2\pi x_{i}))+20+e$	30	[−32, 32]	0
F4	Rastrigin	${\;F}_{5}\left(x\right)=\sum_{i=1}^{d}\left[x_{i}^{2}-10\times \cos\left(2\times \pi \times x_{i}\right)+10\right]$	0	[−5.12, 5.12]	$F_{5}$

**Table 4 diagnostics-13-02454-t004:** The results of the comparison of the suggested amended competitive search optimizer with the investigated optimizers.

Algorithm	Index	Sphere	Rastrigin	Ackley	Rosenbrock
WOA [51]	Average	562.128	232.169	73.254	53.624
	Standard deviation	245.154	94.588	56.642	23.251
HHO [52]	Average	435. 876	145.364	22.374	15.627
	Standard deviation	201.563	81.824	11.412	7.537
FOA [53]	Average	364.529	73.0101	5.0524	3.261
	Standard deviation	1835.624	51.0264	2.0624	2.041
ACSO	Average	109.542	1.3647 × 10^−5^	2.097 × 10^−6^	0.951 × 10^−2^
	Standard deviation	96.637	0.038 × 10^−5^	1.052 × 10^−6^	0.121 × 10^−2^

**Table 5 diagnostics-13-02454-t005:** Comparison assessment of the suggested technique against the others.

Method	Performance Metric			
	Accuracy	Precision	Recall	F1-Score
GLCM [46]	82.44	84.73	82.44	86.25
weight balancing [47]	78.62	80.91	67.93	68.70
SVM [48]	82.44	83.96	81.68	84.73
quadratic discriminant analysis [52]	74.81	74.81	61.83	71.75
mobile microscopy [51]	78.62	75.57	78.62	75.57
transfer learning [50]	81.67	80.91	75.57	78.62
deep method [49]	82.44	80.91	72.15	75.57
proposed method	94.65	93.89	82.44	86.27

## Data Availability

Not applicable.

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
