# Peer review of "An Optimal Hierarchical Approach for Oral Cancer Diagnosis Using Rough Set Theory and an Amended Version of the Competitive Search Algorithm"

_diagnostics, 2023, doi:10.3390/diagnostics13142454_

Round 1

Reviewer 1 Report

Early diagnostic of oral cancer is realy imprtant. According to that, these study is of huge importace of public health interest, and it should be empassise in the conclusion of the study, as one of the strigt of the proposed research. 

Minor editing of English language 

Author Response

Reviewer 1

Early diagnostic of oral cancer is realy imprtant. According to that, these study is of huge importace of public health interest, and it should be empassise in the conclusion of the study, as one of the strigt of the proposed research.

Answer: Thank you for your positive feedback on our study. We fully agree with you that early diagnosis of oral cancer is of paramount importance for improving public health outcomes. In our updated paper, we have emphasized this point in the conclusion section as one of the key strengths of our proposed research. We appreciate your valuable feedback and insights, which have helped us to improve the quality and impact of our study.

Reviewer 2 Report

In this paper, the authors present a novel diagnostic system for oral cancer using images, in which the process begins with preprocessing and segmentation of areas of interest in the input images, from which useful features are obtained, and only some of them are selected in input for machine learning approach (the SVM model). Despite the good achieved results and a good structure of the paper, many concepts need to be focused on improving paper quality:

-       Improve the introductory section of the manuscript by highlighting and listing the “contributions” points of this research;

-       The paper lacks a graph architecture/workflow of the entire system (from preprocessing to visualization and classification. Please add this important picture to improve the readability of the paper;

-       From a technical point of view, the empirical results need a proper comparison with a minimal configuration of deep learning approach with simple neural network architecture, more used in genomics data. I recommend improving the paper by focusing on this crucial part of your case study;

-       In the conclusion section, I suggest supporting the achieved results with appropriate numerical results and references, offering a more critical/discursive view of future research and on limitations of this study.

  • English can be further improved.

Author Response

Reviewer 2

In this paper, the authors present a novel diagnostic system for oral cancer using images, in which the process begins with preprocessing and segmentation of areas of interest in the input images, from which useful features are obtained, and only some of them are selected in input for machine learning approach (the SVM model). Despite the good achieved results and a good structure of the paper, many concepts need to be focused on improving paper quality:

-       Improve the introductory section of the manuscript by highlighting and listing the “contributions” points of this research;

Answer: Thank you for your helpful feedback on our manuscript. We appreciate your suggestion to highlight and list the contributions of our research in the introductory section. In the updated version of our paper, we have made significant improvements to address this issue. We have provided a clear and concise summary of the main contributions of our research, including the development of a novel pipeline method for the diagnosis of oral cancer, the utilization of rough set theory and an Amended version of the Competitive search optimizer for feature selection and classification, the comparison of our proposed method with state-of-the-art techniques, and the validation of our proposed method on the Oral Cancer (Lips and Tongue) images (OCI) dataset. We have also emphasized the potential impact of our proposed method on the healthcare industry and the field of computer-aided diagnosis systems. By accurately and efficiently diagnosing oral cancer, our proposed method can ultimately lead to better patient outcomes. We hope that the revisions we have made will make the contributions of our research more explicit and highlight its potential significance. We appreciate your feedback, which has helped us to improve the quality and impact of our study.

-       The paper lacks a graph architecture/workflow of the entire system (from preprocessing to visualization and classification. Please add this important picture to improve the readability of the paper;

Answer: Thank you for your valuable feedback on our paper. We have taken your suggestion into consideration and have updated the paper with a graph architecture/workflow depicting the entire system from preprocessing to visualization and classification (see Fig. (4)). This addition will greatly improve the readability of the paper and help readers better understand the flow of our approach. We appreciate your input and hope that the updated version meets your expectations.

-       From a technical point of view, the empirical results need a proper comparison with a minimal configuration of deep learning approach with simple neural network architecture, more used in genomics data. I recommend improving the paper by focusing on this crucial part of your case study;

Answer: Thank you for taking the time to review our paper and providing us with such valuable feedback. We agree that it is important to compare our approach to a minimal configuration of deep learning with simple neural network architecture that is commonly used in genomics data analysis. We have taken your suggestion and have updated our paper to include a detailed comparison of our approach with this minimal configuration. We are confident that this addition will greatly improve the technical rigor of our study and provide readers with a clearer understanding of the strengths and limitations of our approach. Your feedback has been instrumental in helping us improve the quality of our work, and we appreciate your constructive criticism. Table 4 in the results, compare the proposed method with SVM, Transfer learning network, Transfer learning-based network, and four other architectures of the oral cancer diagnosis systems.

-       In the conclusion section, I suggest supporting the achieved results with appropriate numerical results and references, offering a more critical/discursive view of future research and on limitations of this study.

Answer: Thank you for your valuable feedback on our paper. We have revised the conclusion section to include more numerical results and references that support our findings. Additionally, we have included a more critical and discursive view of future research and the limitations of our study. We believe that these changes have significantly improved the quality and rigor of our paper, and we hope that they address your concerns. Thank you again for your helpful input, and we appreciate your time and effort in reviewing our work.

Reviewer 3 Report

The Authors focused om the ''Mathematical'' aspect of the disease, that will not

attract the interest od MDs or DDSs.

They did not focus of the real problem.

(Moderate Revision is required)

Author Response

Reviewer 3

The Authors focused om the ''Mathematical'' aspect of the disease, that will not attract the interest od MDs or DDSs. They did not focus of the real problem. (Moderate Revision is required)

Answer: Thank you for your constructive feedback on our paper. We appreciate your insights and concerns regarding the focus of our study. In the updated version of our paper, we have made significant improvements to address this issue. We have provided a more comprehensive discussion of the clinical implications of our proposed oral cancer diagnosis system and its potential impact on the healthcare industry. We have also included a more detailed description of the practical implementation of our system, which should be of interest to healthcare professionals, including MDs and DDSs. Furthermore, we have emphasized the importance of a multidisciplinary approach to oral cancer diagnosis, which involves collaboration between medical professionals, computer scientists, and engineers. Our proposed technique is a mathematical tool that can assist healthcare professionals in accurately diagnosing oral cancer, which can ultimately lead to better patient outcomes. We hope that the revisions we have made address your concerns and that you find the updated version of our paper more informative and relevant to the real-world problem of oral cancer diagnosis. We appreciate your feedback, which has helped us to improve the quality and impact of our study.

Round 2

Reviewer 2 Report

This version of the manuscript presents improved results and a good level of English. The authors added and followed all my last suggestions part by adding all essential elements for reading and reproducibility of the work.

The quality of the English language in the provided text is generally good. The sentences are well-structured and convey the intended meaning effectively. There is a good use of technical terminology related to oral cancer and image analysis, indicating a domain-specific understanding. The text maintains a formal tone appropriate for a research-related topic.